# Negative regulation of angiogenesis and the MAPK pathway may be a shared biological pathway between IS and epilepsy

Longhui Fu[1][◉], Beibei Yu[1][◉], Boqiang Lv[1], Yunze Tian[1], Yongfeng Zhang[1], Huangtao Chen[1], Shijie Yang[1], Yutian Hu[1], Pengyu Ren[1]*, Jianzhong Li[2]*, Shouping Gong[1,3]*

1 Department of Neurourgery, Second Affiliated Hospital of Xi'an Jiao Tong University, Xi'an, China,
2 Department of Thoracic Surgery, Second Affiliated Hospital of Xi'an Jiao Tong University, Xi'an, China,
3 Xi'an Medical University, Xi'an, China

◉ These authors contributed equally to this work.
* renpengyu.xjut@mail.xjtu.edu.cn (PR); jianzhong-0520@163.com (JL); shpingg@126.com (SG)

## Abstract

Ischemia stroke and epilepsy are two neurological diseases that have significant patient and societal burden, with similar symptoms of neurological deficits. However, the underlying mechanism of their co-morbidity are still unclear. In this study, we performed a combined analysis of six gene expression profiles (GSE58294, GSE22255, GSE143272, GSE88723, GSE163654, and GSE174574) to reveal the common mechanisms of IS and epilepsy. In the mouse datasets, 74 genes were co-upregulated and 7 genes were co-downregulated in the stroke and epilepsy groups. Further analysis revealed that the co-expressed differentially expressed genes (DEGs) were involved in negative regulation of angiogenesis and the MAPK signaling pathway, and this was verified by Gene Set Enrichment Analysis of human datasets and single cell RNA sequence of middle cerebral artery occlusion mice. In addition, combining DEGs of human and mouse, PTGS2, TMCC3, KCNJ2, and GADD45B were identified as cross species conserved hub genes. Meanwhile, molecular docking results revealed that trichostatin A and valproic acid may be potential therapeutic drugs. In conclusion, to our best knowledge, this study conducted the first comorbidity analysis of epilepsy and ischemic stroke to identify the potential common pathogenic mechanisms and drugs. The findings may provide an important reference for the further studies on post-stroke epilepsy.

## 1. Introduction

Stroke, also known as cerebrovascular accident, is the main cause of death (64.4% of total deaths) and disability (42.2% of disability-adjusted life years) worldwide in neurological disorders [1]. Stroke is divided into ischemic stroke (IS), hemorrhagic stroke, and transient ischemic attack [2]. Post-stroke epilepsy (PSE) is one of the common complications of IS, occurring in 87% of strokes [3]. Moreover, PSE increases the risk of death in stroke patients and leads to

**Data Availability Statement:** The datasets GSE143272, GSE58294, GSE22255, GSE88723, GSE163654, and GSE174574 for this study can be

found in the Gene Expression Omnibus (http://www.ncbi.nlm.nih.gov/geo/).

**Funding:** This work was supported by Xi 'an Science and Technology Plan (21YXYJ0116), the Key Research And Development Project of Shaanxi Province (Grant No.2022ZDLSF04-01, and No.2019KW-071), The National Natural Science Foundation of China (Grant No. 81971766, and Grant No. 81903268), and China Postdoctoral Science Foundation (No.2021M692577). The funders had no role in study design, data collection and analysis, decision to publish, or preparation of the manuscript.

**Competing interests:** The authors have declared that no competing interests exist.

**Abbreviations:** DEGs, differentially expressed genes; IS, ischemic stroke; PSE, post-stroke epilepsy; MCAO, middle cerebral artery occlusion; scRNA-seq, single-cell RNA sequencing; GSEA, gene set enrichment analysis; BBB, blood-brain barrier; GO, Gene Ontology; KEGG, Kyoto Encyclopedia of Genes and Genomes.

neurological dysfunction, poor prognosis, prolonged hospitalization, and delayed rehabilitation of stroke survivors [4]. With the aging of the population and the development of medical and health services, the number of PSE patients is increasing. Consequently, PSE imposes a heavy burden to social medical care [5]. Therefore, further understanding the correlation between IS and epilepsy will significantly improve the quality of life of PSE patients.

Epilepsy and IS share similar symptoms of neurological deficits such as spasticity, but research on the common pathogenic mechanisms of these two diseases remains limited. From a pathophysiological perspective, IS most commonly occurs due to vascular disorder, which is usually caused by the occlusion of the arteries that results in reduced blood flow, and rarely due to occlusion of the cerebral veins or venous sinuses [2]. Correspondingly, neurovascular dysfunction such as increased blood-brain barrier (BBB) permeability is an important pathogenesis of epilepsy [6]. In particular, the BBB limits the paracellular diffusion of ions (such as Na +, K +, and Cl -), macromolecules, and polar solutes through tight and adherens junctions between brain microvessel endothelial cells, but permits oxygen and carbon dioxide [7]. However, larger molecules can passively cross the BBB when its permeability is raised, which under pathological circumstances prevents precise cellular and homeostatic connections in favor of aberrant neuronal activity or convulsions [6]. IS has also been recognized as a major risk factor for epileptic seizures in adulthood, as the BBB has been damaged [8]. However, research on the treatment of PSE mainly focuses on antiepileptic drugs which are not recommended as first-line preventive treatment drug [9]. Accordingly, exploring the co-pathogenesis of IS and epilepsy may provide new insights for the treatment of PSE.

Transcriptome sequencing technology is a recent technology that has addressed the issue of insufficient sample size of PSE patients and facilitated the research for co-pathogenesis of IS and Epilepsy. In this study, we identified the differentially expressed genes (DEGs) of IS and epilepsy in the mouse model and found a common biological pathway. Subsequently, cell localization of DEGs in the single cell dataset of mouse middle cerebral artery occlusion (MCAO) model was further carried out. After crossing with DEGs of IS and epilepsy in human peripheral blood and mouse tissue, drug prediction and molecular docking were performed in hub genes. The common regulatory pathway and hub gene of IS and epilepsy, as well as potential therapeutic drugs, were identified through bioinformatics. A flow-diagram illustrating our research process is presented in Fig 1.

## 2. Materials and methods

### 2.1 Datasets acquisition

Gene expression profiles were obtained from the Gene Expression Omnibus (http://www.ncbi.nlm.nih.gov/geo/), an open-source database. As shown in Table 1, gene expression matrixes of 217 human and 18 mouse were collected in six datasets (GSE58294, GSE22255, GSE143272, GSE88723, GSE163654, and GSE174574). In the Bulk RNA sequencing (Bulk RNA-seq), the expression matrices of 3 sham mice and 3 MCAO mice were collected from GSE163654 and 3 sham mice and 3 kainate-induce seizures mice from GSE88723. In addition, single-cell RNA sequencing (scRNA-seq) data of 3 sham mice and 3 MCAO mice were obtained from GSE174574. Finally, expression matrices were extracted from GSE58294 and GSE22255 for 43 healthy individuals and 89 IS patients and from GSE143272 for 51 healthy individuals and 34 epilepsy patients.

### 2.2 Differential expression analysis

R software (v4.1.0, R Foundation, Vienna, Austria) was used for all analysis and visualization in this study. All the raw matrices of Bulk RNA-seq were combined with the corresponding

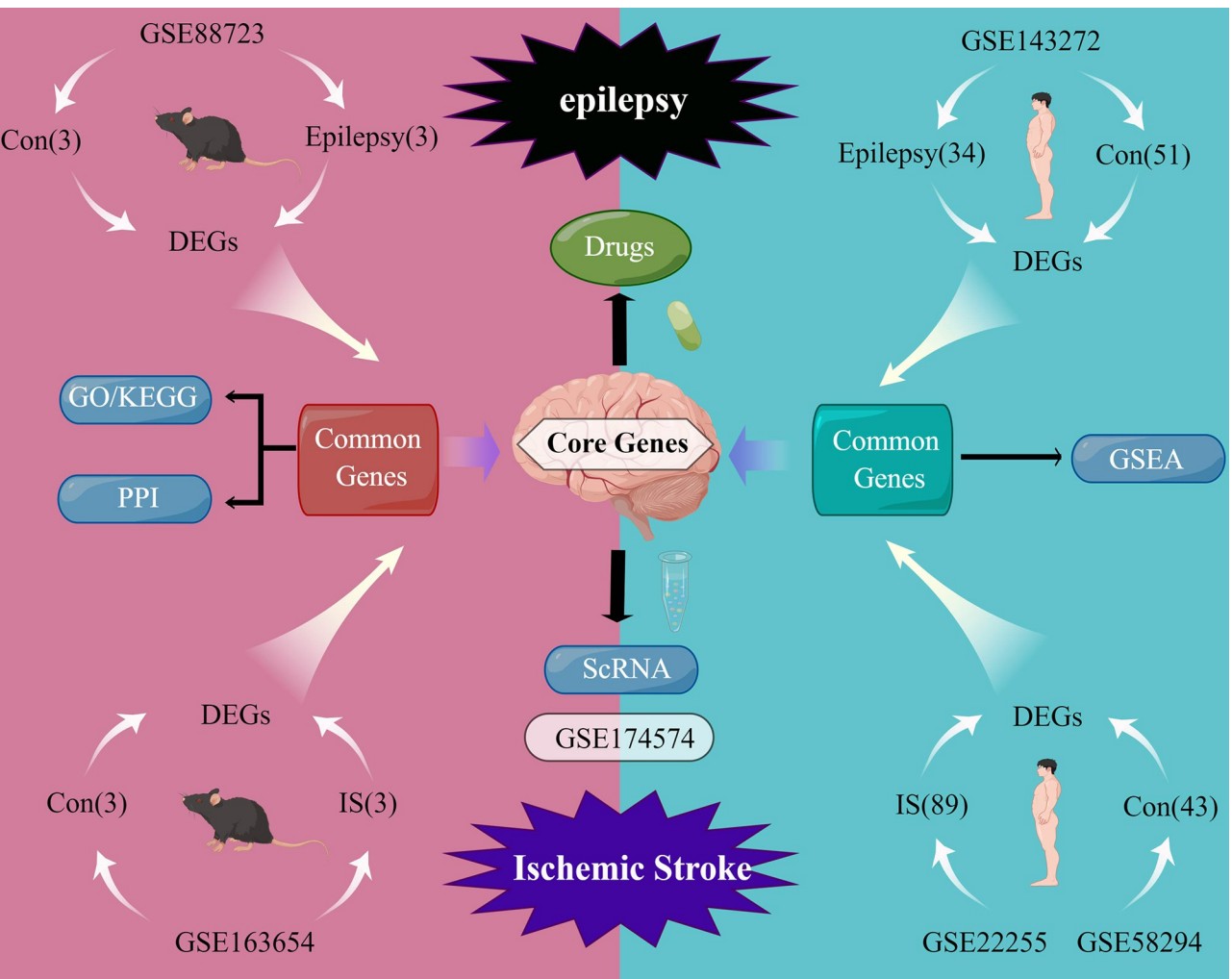

**Fig 1. The flowchart of data preparation and analysis.**

RNA probes after normalization to form a new profiling matrix, using the limma package. The criterion for screening the DEGs was |log2 (fold change)| > 0.3 and P value less than 0.05. Subsequently, a heatmap and volcano plot were created using the heatmap and ggplot2 packages, respectively. The annular heatmap was created using the OmicCircos package. Lastly, Venn analysis was performed using Venn the diagram website (http://www.bioinformatics.com.cn/).

**Table 1. Detailed information of the gene expression matrixes and platform.**

| GEO Dataset | Platform | Disease | Species | Control | experimental | Country |
|---|---|---|---|---|---|---|
| GSE143272 | GPL10558 | Epilepsy | Human | 51 | 34 | India |
| GSE58294 | GPL570 | IS | Human | 69 | 23 | USA |
| GSE22255 | GPL570 | IS | Human | 20 | 20 | Portugal |
| GSE88723 | GPL6096 | Epilepsy | Mouse | 3 | 3 | Germany |
| GSE163654 | GPL16570 | IS | Mouse | 3 | 3 | Canada |
| GSE174574 | GPL21103 | IS | Mouse | 3 | 3 | China |

### 2.3 Protein interaction network and pathway enrichment analysis

The express analysis method of Metascape website (http://metscape.ncibi.org) was performed to analyze protein interaction networks and pathway for DEGs [10]. Pathway enrichment analysis was conducted using the ClueGO v2.5.9 app of Cytoscape v3.8.2 software with P value < 0.05. Moreover, MCODE v1.6.1 plugin was used to identified highly interconnected clusters in DEGs. Gene set enrichment analysis (GSEA) was performed using the GSEA software (http://www.broadinstitute.org/gsea/). After screening for pathways with a P value < 0.05, gseaplot2 package was applied to plot the GSEA pathway enrichment graph.

### 2.4 Drug screening and molecular docking

Based on the hub DEGs, our study predicted FDA-approved drugs and experimental compounds included in DSigDB database (http://tanlab.ucdenver.edu/DSigDB). The screening criterion was an adjusted P value < 0.05 and combined score>3000, and the ranking was based on the comprehensive score. Thereafter, three PDBQT files were we specified: proteins rigid of DEGs, flexible, and ligands of drugs with the highest score. Ultimately, the AutoDock 4 software was used to conduct molecular docking.

### 2.5 ScRNA-seq data processing

Before data analysis, scRNA-seq data were analyzed using the Seurat package for unsupervised graph-based clustering. For the screening criteria, only cells with 500–6,000 UMI (unique molecular identifiers) and <35% of mitochondrial genes were included; otherwise, they were considered as low-quality cells and excluded from subsequent analysis. The quality-controlled data were normalized using the Normalize Data function, and 2000 highly variable genes were selected using the Find Variable Features function. The data were integrated using the mutual principal component analysis function of Seurat. The first 20 principal components were selected for visualization of dimensionality reduction using t-distributed stochastic neighbor embedding. The cell types were identified by the marker genes, using SingleR package. And the expression of hub genes were visualized using FeaturePlot and VlnPlot.

## 3. Results

### 3.1 Identification and function enrichment of hub genes in mouse Bulk RNA-seq

GSE163654 and GSE88723 were analyzed, and heatmaps were plotted based on the differential genes between the experimental and sham group mice (Fig 2A and 2B). Among them, 324 genes were upregulated and 299 genes were downregulated in the IS group, and 724 genes were upregulated and 592 genes were downregulated in the epilepsy group (Fig 2C). Venn diagrams showed that 74 genes were co-upregulated and 7 genes were co-downregulated in the stroke and epilepsy groups. After analysis of these 81 genes in the Metascape website, these hub genes were highly enriched in positively regulating cell death, skeletal muscle cell differentiation, and positively regulating smooth muscle cell proliferation and the MAPK signaling pathway (Fig 3A). In the protein interaction network, Cluster 1 (Fosb, Fos, Fosl2, Jun, Junb, Ccl7, Nfil3, and Phf21b) was enriched in transcription by RNA polymerase II; Cluster 3 (Adamts1, Adamts9, and Thbs1) was enriched in the negative regulation of angiogenesis; and Cluster 4 (Dusp1, Dusp6, Vcl) was enriched in the MAPK signaling pathway (Fig 3B). In addition, we also performed GSEA analysis of human Bulk RNA-seq separately, and the results indicated that angiogenesis as well as the MAPK signaling pathway are important common

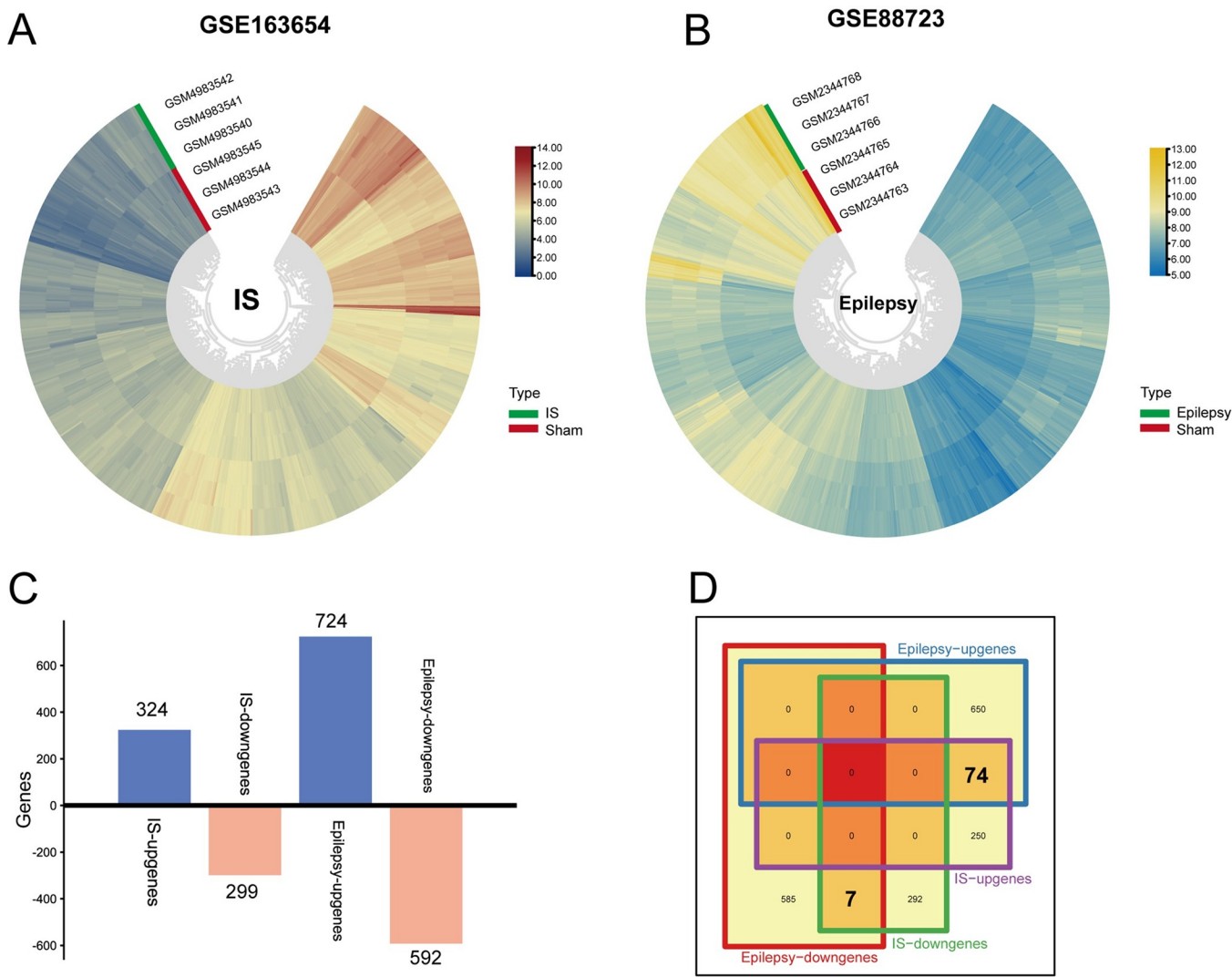

**Fig 2. Discovery of DEGs in mouse Bulk RNA-seq.** (A) Cluster heatmap for DEGs in GSE163654 dataset. Red represents high gene expression and blue represents low expression. (B) Cluster heatmap for DEGs in GSE88723 dataset. Yellow represents high gene expression and blue represents low expression. (C) Number of DEGs up or down regulated in IS and epilepsy. (D) Venn diagrams of overlap in co-DEGs between IS and epilepsy.

pathways between IS and epilepsy (Fig 3C and 3D). The location where the 81 genes participate in MAPK pathway are shown in S1 Fig.

## 3.2 Identification of hub genes expressed in IS by scRNA-seq

To further obtain the cellular localization of key pathway genes, we selected Thbs1, Adamts9, Adamts1, Dusp6, Dusp1, and Vcl in Cluster 3 and Cluster 4 as key genes for scRNA-seq analysis in the MCAO mouse model. Base on the marker genes, uniform manifold approximation and projection analysis of the MCAO group and the sham group identified nine cell clusters, including astrocytes, endothelial cells, epithelial cells, fibroblasts, granulocytes, microglia, monocytes, NK cells and oligodendrocytes (Fig 4A–4C). The expression of the six genes in various cell clusters and groups was further analyzed, and the results were visualized in uniform manifold approximation and projection (Fig 4D and 4E). Thbs1 was mainly expressed in endothelial cells, microglia, monocytes, and astrocytes, and its expression was higher in

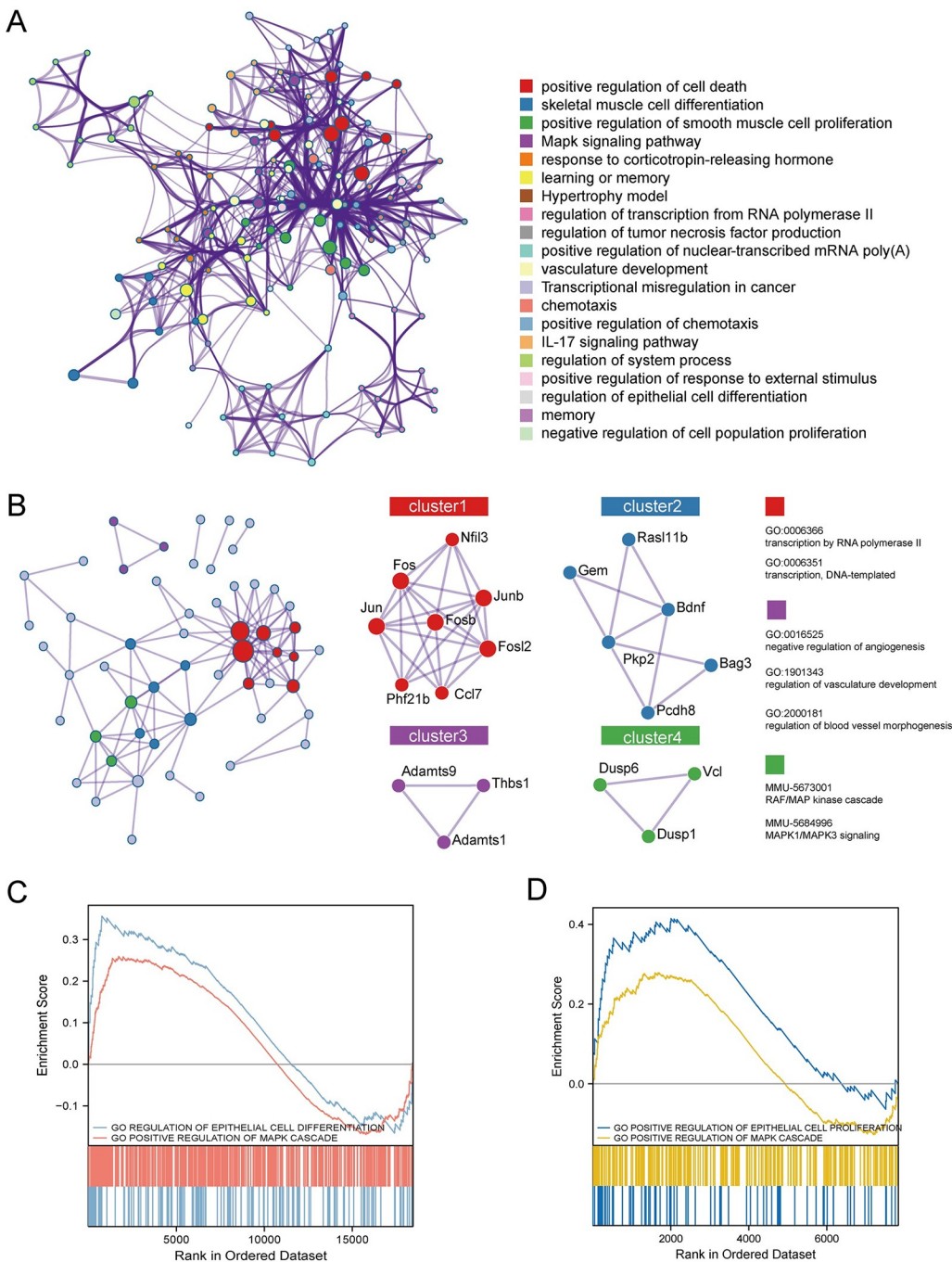

**Fig 3. Identification and function enrichment of hub genes in mouse Bulk RNA-seq.** (A) Pathway analysis and protein interaction network of DEGs. The vertical axis from top to bottom represents the P-value from small to large. The color of the proteins corresponds to the pathway. (B) The top four clusters and their corresponding protein. (C) GSEA analysis of human Bulk RNA-seq in IS. (D) GSEA analysis of human Bulk RNA-seq in epilepsy.

monocytes. Adamts9 was significantly expressed mainly in endothelial cells. Adamts1 and Dusp6 were significantly expressed in endothelial cells, microglia, monocytes, epithelial cells and astrocytes, with Adamts1 expression being significantly higher in endothelial cells and microglia, and Dusp6 expression being significantly higher in endothelial cells, microglia, and

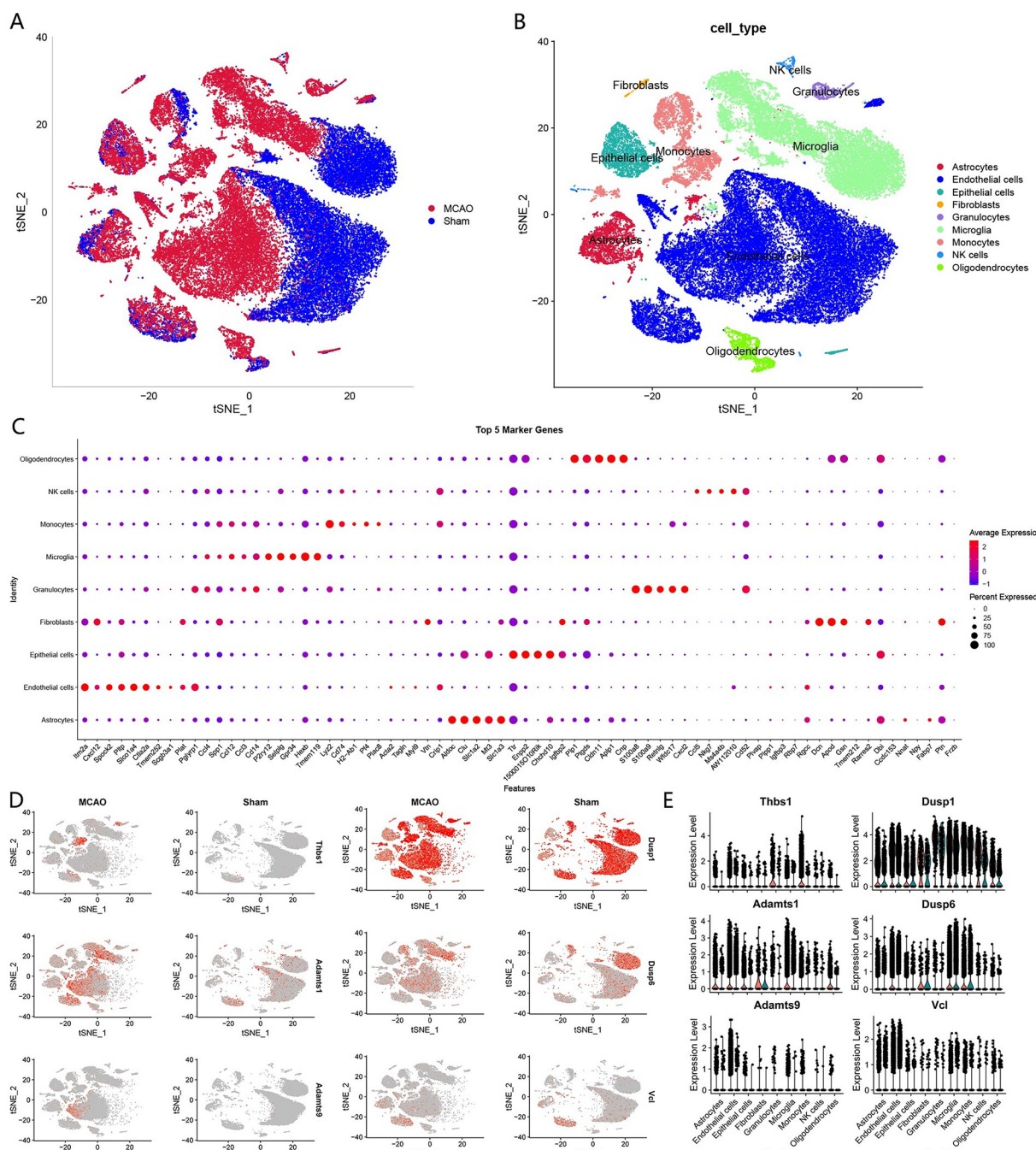

**Fig 4. The scRNA-seq reveals the expression of hub genes in mouse brain.** (A) Cluster analysis of scRNA-seq in GSE174574 dataset. Red represents the cells in the MCAO group and blue represents the cells in the Sham group. (B) Identification of cell clusters obtained in (A). Different colors represent different cell clusters, and a total of 9 cell clusters are identified. (C) Top 5 marker genes of 9 cell types. (D) Distribution of hub genes expression in different cell clusters and groups. Red represents the high expression of genes. (E) Quantified expression of hub genes in different cell clusters and groups. The red on the left shows expression of hub genes in MCAO group, and the green on the right in Sham group.

monocytes. Dusp1 was widely and significantly expressed in all the 9 types of cells. Vcl was significantly expressed mainly in endothelial cells and astrocytes. In addition, the expression levels of most hub genes in the MCAO group were higher than those in the Sham group,

especially of Thbs1, Adamts9 and Adamts1. Collectively, all of these key pathway genes are expressed to varying degrees in endothelial cells, which was consistent with our results of pathway enrichment in Bulk RNA-seq.

### 3.3 Identification of cell subtypes within endothelial cells

In order to discern the intracellular subtypes associated with IS within endothelial cells, we partitioned the endothelial cells into seven distinct clusters, designated as EC0 through EC6 (Fig 5A). Following this, we conducted a comparative analysis of the various subtypes present in both the MCAO and Sham cohorts (Fig 5B). Our observations revealed that the Sham

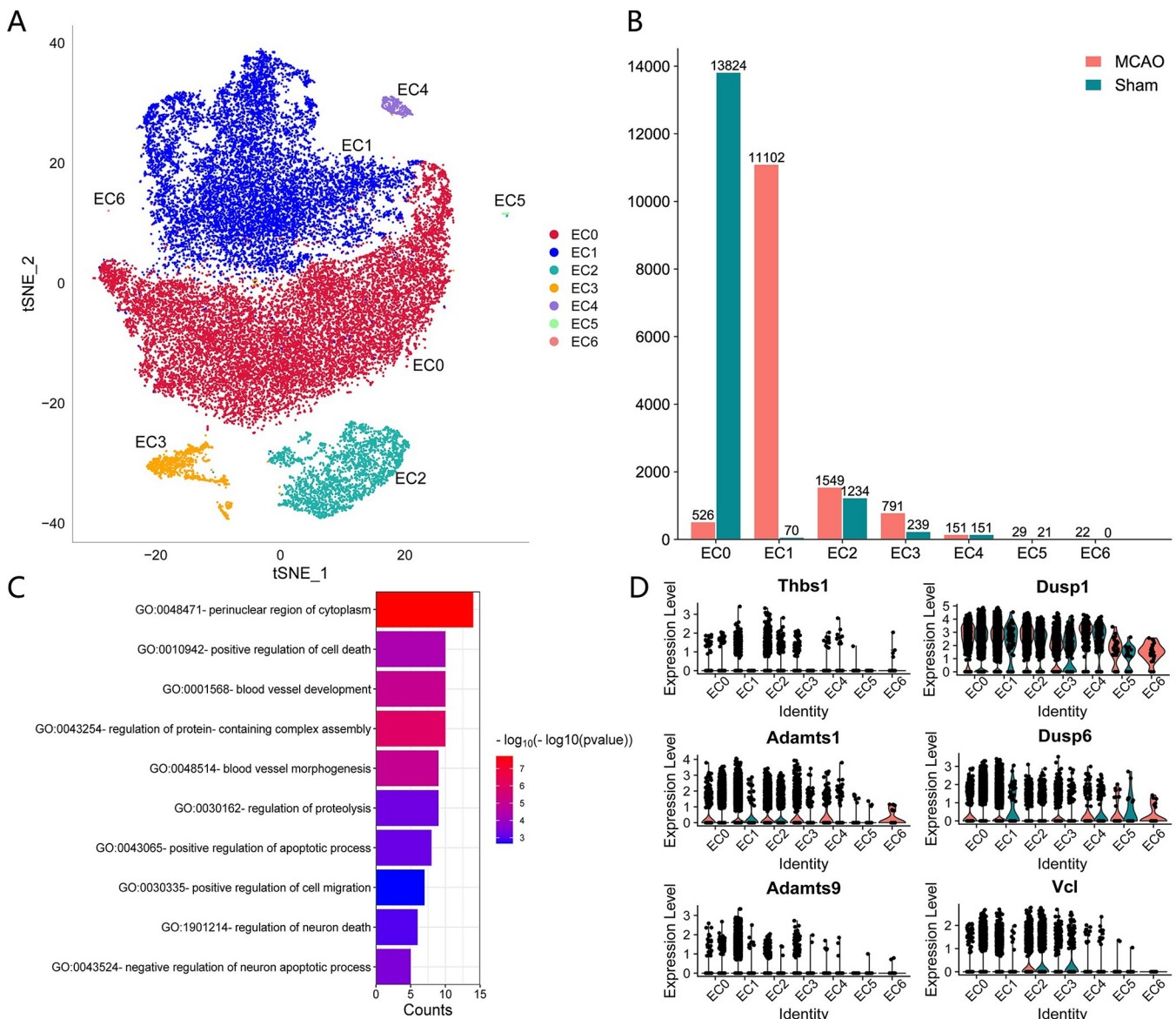

**Fig 5. Identification of cell subtype within endothelial cells.** (A) Identification of subtypes of endothelial cells. The seven subtypes were named EC0-EC6. (B) The number distribution of different subtypes of cells in MCAO group and Sham group. The number of cells identified on the vertical axis and the cell subtypes on the horizontal axis. (C) GO/KEGGG functional enrichment of the 69 marker genes based on EC1. (D) Quantified expression of hub genes in different subtypes and groups.

group predominantly comprised EC0 cells with a paucity of EC1 cells, whereas the MCAO group exhibited a substantial upsurge in the number of EC1 cells with an almost negligible presence of EC0 cells. To further elucidate the functional role of the EC1 (Ctla2a+, Tmem252 +), we executed Gene Ontology (GO) and Kyoto Encyclopedia of Genes and Genomes (KEGG) enrichment analyses on the 69 marker genes correlated with EC1, with the outcomes delineated in Fig 5C. These genes were highly enriched in perinuclear region of cytoplasm, positive regulation of cell death, blood vessel development, etc. Subsequently, we explored the expression patterns of pivotal genes across diverse endothelial cell subtypes (Fig 5D). Our results indicate that the expression of cardinal genes within the EC1 subtype was markedly elevated in the MCAO group as compared to the Sham group, shedding light on the potential role of EC1 in the context of ischemic stroke.

### 3.4 Identification and function enrichment of hub genes in human Bulk RNA-seq

After combining GSE22255 and GSE58294, 2272 differential genes were found in IS patients, of which 980 genes were upregulated and 1292 genes were downregulated (Fig 6A and 6C). In the analysis of GSE143272, 338 differential genes were found, of which 162 genes were upregulated and 176 genes were downregulated in epilepsy patients (Fig 6B and 6D). Venn analysis of the DEGs in IS and epilepsy patients revealed 62 hub genes (Fig 6E). Finally, the expression matrices of these 62 hub genes in IS and epilepsy were displayed in heatmaps (Fig 6F and 6G). GO enrichment analysis revealed the biological pathways in which these 62 hub genes were involved and mainly included acute inflammatory response, specific granules, neutrophil-mediated immunity, primary lysosomes, and negative regulation of intrinsic apoptotic signaling pathways (Fig 7A). Subsequent Kyoto Encyclopedia of Genes and Genomes analysis suggested that these genes may be involved in thyroid cancer, NF-κB signaling pathway, transcriptional misregulation in cancer, IL-17 signaling pathway, and TNF signaling pathway (Fig 7B).

### 3.5 Conservative genetic screening and molecular docking

There were four intersecting genes between 81 DEGs in the mouse data set and 62 DEGs in the human data set (Fig 8A). Subsequently, these four genes were localized; they were PTGS2 on human chromosome 1, TMCC3 on human chromosome 12, KCNJ2 on human chromosome 17, and GADD45B on chromosome 19 (Fig 8B). Using the DSigDB database, we identified small molecule compounds that may bind to PTGS2, TMCC3, KCNJ2, and GADD45B. Accordingly, the protein structures of PTGS2, TMCC3, KCNJ2, and GADD45B in the PDB database were investigated. Further, the small molecule compounds with the lowest binding energy were identified for potential target drugs. Among them, trichostatin A could bind to TMCC3, KCNJ2, and GADD45B, and valproic acid could bind to KCNJ2 and GADD45B (Fig 8C and 8D).

## 4. Discussion

Epilepsy and IS are common neurological disorders in the elderly, and IS is also a cause of seizures in the elderly and epilepsy is a common complication after IS [2, 8]. The incidence of IS is increasing annually; accordingly, the co-pathogenesis of IS and epilepsy is gaining increasing attention [11]. To explore the co-pathogenesis of IS and epilepsy, this study conducted a joint analysis of human samples and mouse samples through bioinformatics. The mouse sequencing analysis identified angiogenesis and the MAPK signaling pathway as the common signal pathway involved in IS and epilepsy. Subsequent sc-RNA seq analysis was further carried out for

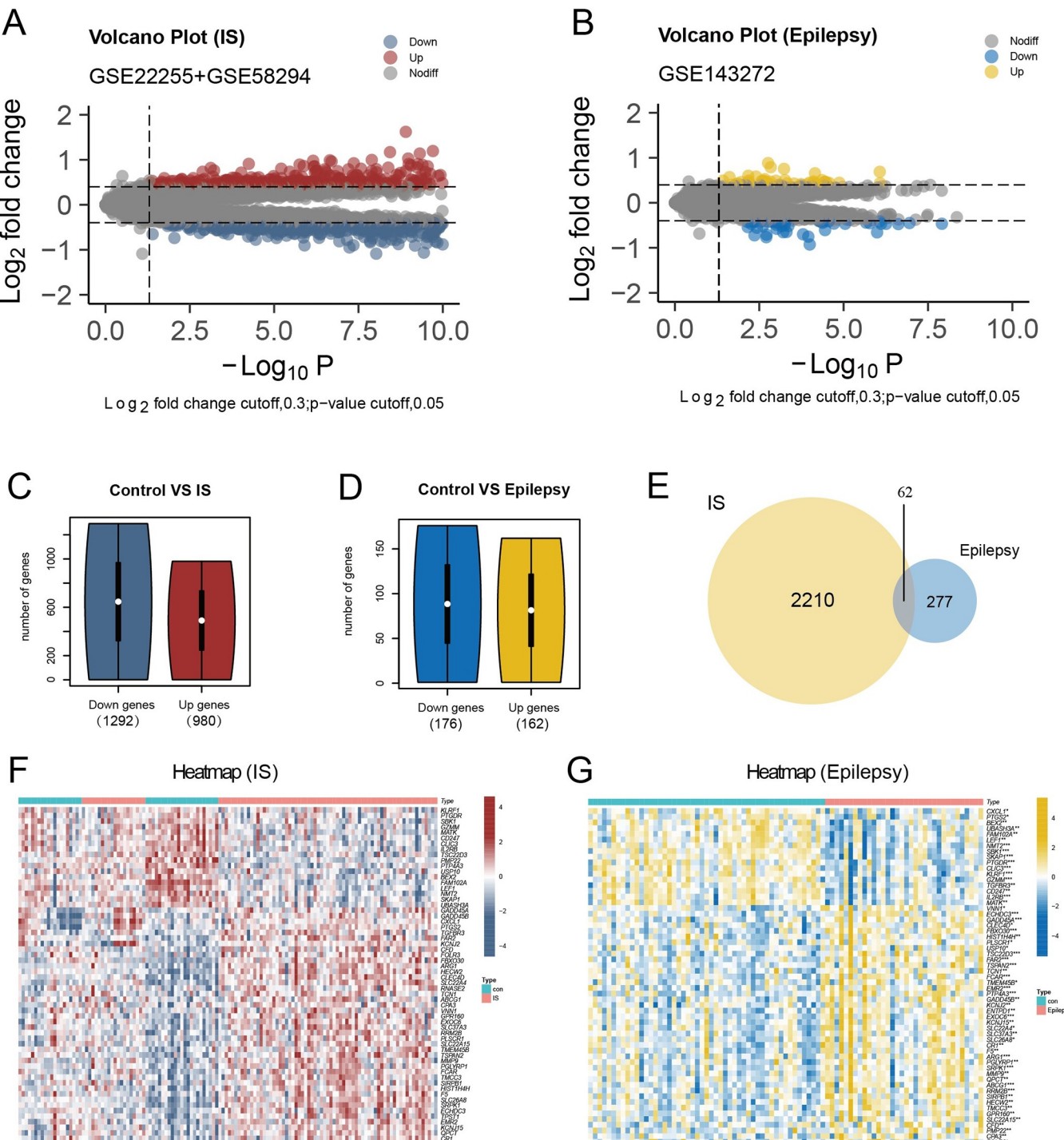

**Fig 6. Discovery of DEGs in human Bulk RNA-seq.** (A) Volcano plot for DEGs in GSE22255 and GSE58294 dataset. (B) Volcano plot for DEGs in GSE143272 dataset. (C) The number of up and down regulated DEGs in (A). (D) The number of up and down regulated DEGs in (B). (E) Venn diagrams of overlap in co-DEGs between IS and epilepsy. (F) Cluster heatmap for DEGs in GSE22255 and GSE58294 dataset. Red represents high gene expression and blue represents low expression. (G) Cluster heatmap for DEGs in GSE143272 dataset. Yellow represents high gene expression and blue represents low expression.

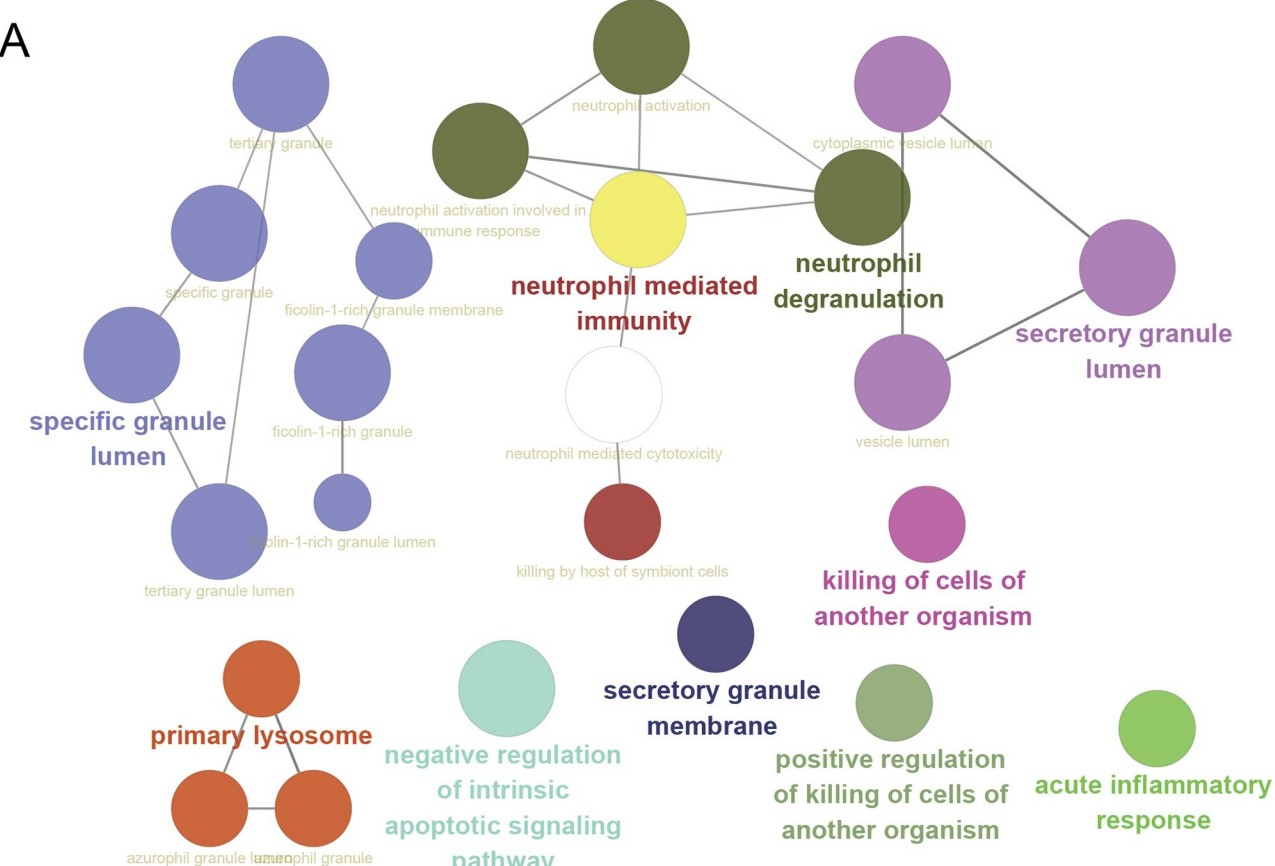

**Fig 7. Function enrichment of DEGs in human Bulk RNA-seq.** (A) The biological pathways involved in DEGs obtained by GO enrichment analysis and the interrelationships between pathways. (B) KEGG pathway analysis of DEGs.

cell localization of hub genes involved in these pathways. The combination of DEGs in mouse samples and human samples identified hub genes involved in IS and epilepsy and potential therapeutic drugs. Along with the findings of functional enrichment analysis of mouse DEGs, the results support that negative regulation of angiogenesis and the MAPK pathway may be a shared biological pathway between IS and epilepsy.

Blood-brain barrier (BBB) alteration is a common pathology of IS, and it serves as the mechanism for secondary damage and the development of hemorrhagic transformation [12].

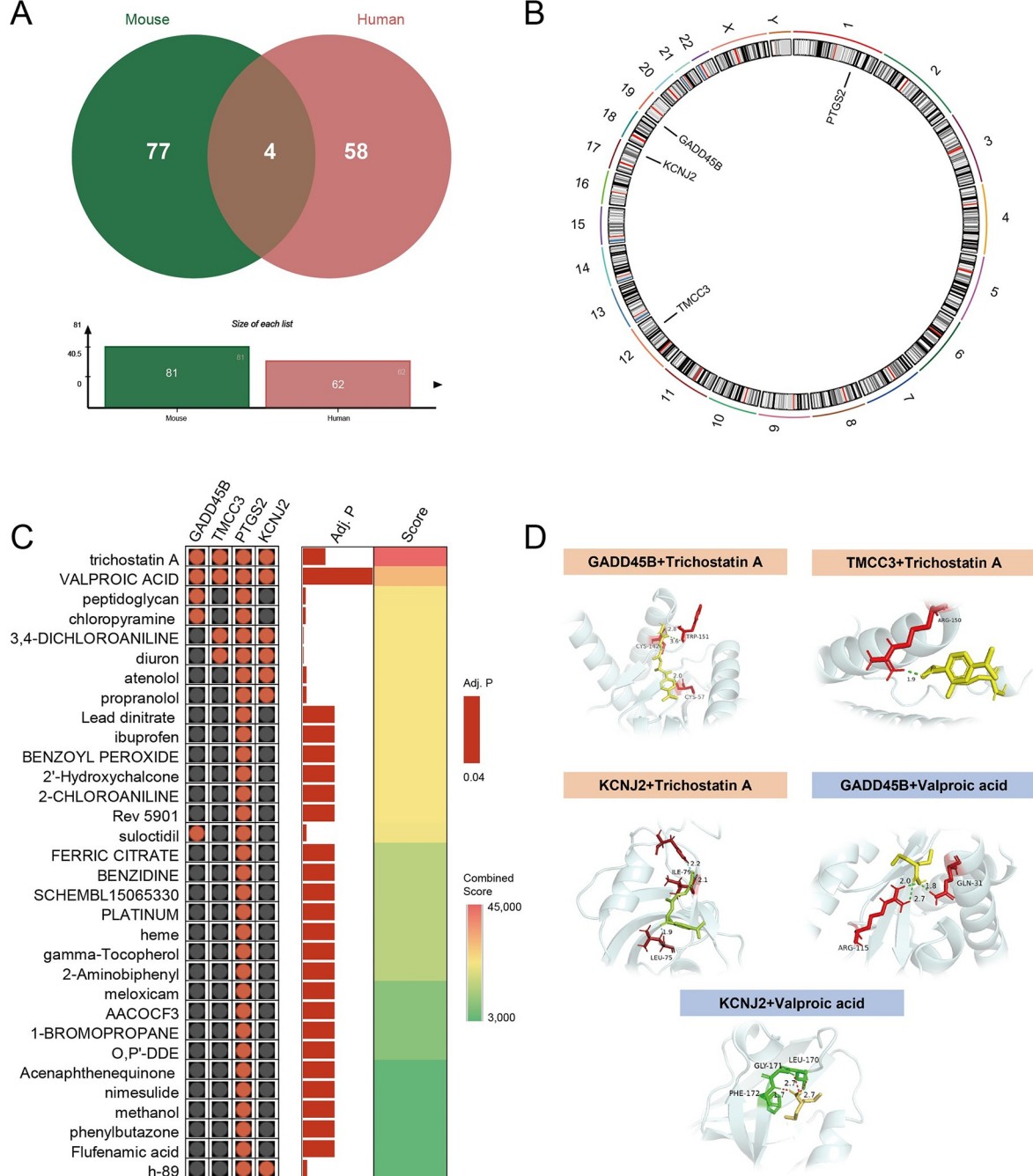

**Fig 8. Conservative genetic screening and molecular docking.** (A) Venn diagrams of overlap in intersecting genes between co-DEGs from the datasets of human origin and of mouse origin. (B) Chromosomal localization of intersecting genes. (C) Small molecule compounds that may bind to PTGS2, TMCC3, KCNJ2 and GADD45B in DSigDB database. (D) Docking simulation of proteins and small molecule compounds.

Meanwhile, epilepsy is clearly induced by permeability alterations and neurovascular unit dysfunction caused by BBB damage [6, 13]. Regulation of angiogenesis can maintain the stability of BBB and can therefore reduce secondary IS damage and decrease PSE seizures [14]. The MAPK signaling pathway is also involved in the etiology of IS and epilepsy. The hub gene of

this pathway, Dusp1, has previously been identified as a hub gene in IS, and consistent findings were obtained in the current study [15]. The occurrence of IS mostly involved an immunological inflammatory response and oxidative stress-related pathways [16, 17]. In addition, several studies have reported that the MAPK and NF-κB pathways jointly participate in oxidative stress and inflammatory response in IS [18, 19]. In the pathogenesis of epilepsy, the MAPK pathway is mainly related to neuroinflammation [20]. Wang et al. recently reported that the MAPK signaling pathway regulates the decrease in phosphorylated myelin basic protein in the hippocampus and may facilitate the prevention of demyelination in epilepsy [21]. In the current study, GSEA enrichment of human samples demonstrated that angiogenesis and MAPK signaling were associated with IS and epilepsy.

Combining the above results, we infer that vascular development and MAPK signaling are important in IS and epilepsy. Thus, six hub pathway genes (Thbs1, Adamts9, Adamts1, Dusp6, Dusp1, Vcl) were analyzed by scRNA-seq to further explore the pathogenesis of the disease. The expression of related genes in endothelial cells, monocytes, microglia, and astrocytes differed significantly between the MCAO and control groups. Notably, these hub genes were all significantly expressed in endothelial cells, and mostly higher in MCAO group, indicating that endothelial cell changes play a role in the pathogenesis of IS. Endothelial cells are an essential component of the BBB, supporting our hypothesis [22]. Endothelial cell inflammation, its interactions with other cells and a variety of cytokines secreted by endothelial cell are partly responsible for the acute thrombotic even advancing and inflammatory response that occurs during IS [23, 24]. Analysis of subtypes of endothelial cells showed that many EC1 appeared after IS, with relatively high hub gene expression levels. These endothelial cells exhibit apoptotic states and are associated with perinuclear region of cytoplasm, positive regulation of cell death, blood vessel development, etc. Notably, these genes are also enriched in regulation of neuron death and negative regulation of neuron apoptotic process, which mean that EC1 may also have a neuroprotective effect. Furthermore, MAPK signaling pathway-related genes were found to be highly expressed in monocytes, microglia, and astrocytes, indicating an immune inflammatory response following IS. Monocyte infiltration is a cause of neural injure after IS, which can be eased when p38 MAPK and NF-κB pathways was restricted [25]. Activation and polarization of microglia and astrocytes are regarded as a double-edged sword for neurological recovery. Microglia can be in a pro-inflammatory condition, which can lead to additional brain injury, meanwhile, they can also release cytokines that are anti-inflammatory and neurotrophic, which aid in stroke healing [26]. Similarly, astrocytes aid in the repair of nerves after a stroke and prevent lesions from spreading, but they inevitably increase brain damage by taking part in the inflammatory response [27]. The regulation of MAPK signaling pathway is beneficial to control the activation of microglia and astrocytes in both the ischemic penumbra of the hippocampus and the hippocampus cerebral cortex following IS [28]. In addition, the monocyte infiltration and activation of microglia usually occur after epileptic states, which can cause damage to the nervous system to some extent and can be controlled when the MAPK pathway is downregulated [20, 29]. However, due to the lack of scRNA-seq of epilepsy, this study was not in-depth here.

In consideration of species conservation, we jointly analyzed the sequencing results of human peripheral blood samples and mouse tissue samples, and four genes (GADD45B, TMCC3, TPGS2, and KCNJ2) were identified as hub genes in IS and epilepsy. Studies show that Gadd45b improves neurogenesis by increasing brain-derived neurotrophic factor and alters the production of pro-inflammatory cytokines in IS [30]. Another study combining scRNA-seq with Bulk RNA-seq also revealed Gadd45b as a specific gene for microglia in the early stages of IS [31]. In addition, Xiao et al. reported that Gadd45b is involved in the regulation of epilepsy through DNA demethylation [32]. Concurrently, KCNJ2 is a potassium

channel-related protein that has been found to selectively regulate the proinflammatory response, resulting in reduced infarct size in MCAO models [33]. However, the involvement of TMCC3 and TPGS2 in IS and epilepsy has not been elucidated.

Based on molecular docking in the current study, two small molecule compounds, trichostatin A and valproic acid, may serve as candidate drugs for IS and epilepsy. A recent study revealed that trichostatin A improved IS by decreasing autophagy and lysosomal dysfunction in ischemic penumbra neurons [34]. Interestingly, George et al. showed that trichostatin A combined with sodium valproate helped improve PSE-induced neurologic impairment in immature mice [35]. Meanwhile, the favorable treatment effect of valproic acid in epilepsy has been well confirmed. Similarly, it has been shown that valproic acid can decrease microglial activation and diminish glial scar formation to relieve IS [36]. Therefore, we hypothesize that trichostatin A combined with valproic acid might be a promising treatment option for PSE.

This study has some limitations. First, although we performed a co-analysis of human and mouse sequencing data to eliminate interspecies differences, the hub genes and drugs must be validated in animal models and patient groups. Second, the main goal of this study was to find the common pathogenesis of IS and epilepsy; therefore, the generalizability of the findings to PSE still needs to be confirmed by sequencing data from relevant models or patients. Finally, owing to the lack of epileptic scRNA-seq data, the expression level of hub genes in epilepsy cells needs to be further explored. Despite these limitations, to our best knowledge, this is the first study to investigate the collective hub genes and biological mechanism of IS and epilepsy. In addition, the small molecule compounds trichostatin A and valproic acid were identified as potential treatment modalities for IS and epilepsy. The study findings provide a novel insight for further research into the management of PSE.

## Supporting information

**S1 Fig. The location of the 81 genes participate in MAPK pathway.**
(PNG)

## Acknowledgments

We would like to thank Editage (www.editage.cn) for English language editing.

## Author Contributions

**Conceptualization:** Shouping Gong.

**Data curation:** Boqiang Lv, Yunze Tian, Yongfeng Zhang, Huangtao Chen, Shijie Yang, Yutian Hu.

**Formal analysis:** Longhui Fu, Beibei Yu, Boqiang Lv, Yunze Tian.

**Writing – original draft:** Longhui Fu, Beibei Yu.

**Writing – review & editing:** Pengyu Ren, Jianzhong Li, Shouping Gong.

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
