## [Decision Letter · Decision Letter 0]

10 Apr 2023

PONE-D-23-03945Negative regulation of angiogenesis and the MAPK pathway may be a shared biological pathway between IS and epilepsyPLOS ONE

Dear Dr. Gong,

Thank you for submitting your manuscript to PLOS ONE. After careful consideration, we feel that it has merit but does not fully meet PLOS ONE’s publication criteria as it currently stands. Therefore, we invite you to submit a revised version of the manuscript that addresses the points raised during the review process.

We look forward to receiving your revised manuscript.

Kind regards,

Yuzhen Xu

Academic Editor

PLOS ONE

“This work was supported by Xi 'an Science and Technology Plan (21YXYJ0116), the Key Research And Development Project of Shaanxi Province (Grant No.2022ZDLSF04-01, and No.2019KW-071), The National Natural Science Foundation of China (Grant No. 81971766, and Grant No. 81903268), and China Postdoctoral Science Foundation (No.2021M692577).”

“The authors declare that the research was conducted in the absence of any commercial or financial relationships that could be construed as a potential conflict of interest.”

Reviewers' comments:

Reviewer's Responses to Questions

**Comments to the Author**

1. Is the manuscript technically sound, and do the data support the conclusions?

Reviewer #1: Yes

Reviewer #2: Yes

2. Has the statistical analysis been performed appropriately and rigorously? 

Reviewer #1: N/A

Reviewer #2: Yes

3. Have the authors made all data underlying the findings in their manuscript fully available?

Reviewer #1: Yes

Reviewer #2: Yes

4. Is the manuscript presented in an intelligible fashion and written in standard English?

Reviewer #1: Yes

Reviewer #2: Yes

5. Review Comments to the Author

Reviewer #1: i accept the manuscript titled "Negative regulation of angiogenesis and the MAPK pathway may be a shared

biological pathway between IS and epilepsy " as the study concluded that the first comorbidity common between

epilepsy and ischemic stroke.

Reviewer #2: In this study, Longhui Fu et al. reported the combined RNA expression analysis of ischemia stroke and epilepsy by using published bulk and single cell sequencing datasets. They found that negative regulation of angiogenesis and the MAPK signaling pathway may functions in a conserved and mechanistic way in the pathogenesis of these two diseases. Further DEG analysis revealed the hub genes, in addition to the potential drugs revealed by prediction of compounds binding. Findings from this study provide novel insights into the common pathogenic mechanisms and drugs underlying ischemia stroke and epilepsy. The expriments are well designed and performed in a high standard. I have some concerns regarding the methods, text format and data analysis.

1. Move the figure legends from the main text to somewhere indicated by the publish policy of Plos One.

2. Authors should consider to change the name of method 2.1 “microarray datasets”, since this part includes bulk and single cell data as well.

3. Detailed parameters used in the software should be disclosed in the method part for reproducibility. For example, in method 2.3, the specific commands used for Metascape and ClueGO, Cytoscape and GSEA should be fully listed. Similar requirements are applied to other packages.

4. The session “Data Availability Statement” should be presented somewhere before “Reference”. Authors may have to rearrange some sessions according to the regulation of Plos One.

5. For scRNA-seq analysis in Figure 4, authors should present the cell markers of each cluster annotated in the main figure, in the format of either dotplots or heatmap. This will make the analysis transparent and readable to the readers.

6. For scRNA-seq analysis in Figure 4, authors should explain the phenotypes that some cell clusters consist of subpopulations characterized by different sample origin (Sham vs Mcao), especially in the clusters of microglia and ¬endothelial cells. The segregation by conditions indicate that these subsets of cells are condition-specific, and may represent a different cell type/status. Authors should address these issues, and consider to reannotate the clusters accordingly. In addition, the resolution of UMAPs is too low to clearly identify cell labels. High resolution images are required all through the figures.

6. PLOS authors have the option to publish the peer review history of their article (what does this mean?). If published, this will include your full peer review and any attached files.

Reviewer #1: No

Reviewer #2: **Yes: **Le Xu

---

## [Author Response · Author response to Decision Letter 0]

5 May 2023

Reviewer #1: i accept the manuscript titled "Negative regulation of angiogenesis and the MAPK pathway may be a shared biological pathway between IS and epilepsy " as the study concluded that the first comorbidity common between epilepsy and ischemic stroke.

Authors’ response: Thank you for your recognition of our research.

Reviewer #2: 

1. Move the figure legends from the main text to somewhere indicated by the publish policy of Plos One.

Authors’ response: Thank you for your comment. According to the “Submission Guidelines” of Plos One: Figure captions are inserted immediately after the first paragraph in which the figure is cited. Figure files are uploaded separately. Therefore, the figure legends will still appear main text. But thanks for your reminding, we have checked the format of the article and found some formatting problems such as the reference format, and revised this.

2. Authors should consider to change the name of method 2.1 “microarray datasets”, since this part includes bulk and single cell data as well.

Authors’ response: Thank you for your comment. It was a lack of thought, and we apologize for that. We have changed the name of method 2.1 “microarray datasets” to “Datasets acquisition”.Details see page 2, line 71.

3. Detailed parameters used in the software should be disclosed in the method part for reproducibility. For example, in method 2.3, the specific commands used for Metascape and ClueGO, Cytoscape and GSEA should be fully listed. Similar requirements are applied to other packages.

Authors’ response: Thank you for your comment. Our description of the method part is really not detailed enough. We have revised this part according to your comments, focusing on the addition of specific implementation methods including detailed parameters. Details see page 3-4, line 91-116.

4. The session “Data Availability Statement” should be presented somewhere before “Reference”. Authors may have to rearrange some sessions according to the regulation of Plos One.

Authors’ response: Thank you for your comment. This was indeed an oversight on our part, and we adjusted the organizational structure of the article. Details see page 2, line 71.

5. For scRNA-seq analysis in Figure 4, authors should present the cell markers of each cluster annotated in the main figure, in the format of either dotplots or heatmap. This will make the analysis transparent and readable to the readers.

Authors’ response: Thank you for your comment. We have added a bubble chart to show the top 5 marker genes for all 9 cell types, and explained it in the “Results”. Details see page 5, line 145-169, and Fig 4C, D.

6. For scRNA-seq analysis in Figure 4, authors should explain the phenotypes that some cell clusters consist of subpopulations characterized by different sample origin (Sham vs Mcao), especially in the clusters of microglia and ¬endothelial cells. The segregation by conditions indicate that these subsets of cells are condition-specific, and may represent a different cell type/status. Authors should address these issues, and consider to reannotate the clusters accordingly. In addition, the resolution of UMAPs is too low to clearly identify cell labels. High resolution images are required all through the figures.

Authors’ response: Thank you for your comment. As a unified annotation method is required for cells in the MCAO group and the Sham group, we did not use an isolated method to annotate cells in the two groups separately. But we also recognize that the lack of group-based discussion is unconvincing in this section. After thinking, we made some modifications and showed the expression of hub genes based on MCAO and Sham groups respectively, and explained it in the “Results”. We also have some discussion about the subtypes of endothelial cells, as we think they play an important role in both diseases. In addition, we have re-provided TIF files with higher resolution. Details see page 5, line 170-189; page 7-8, line 265-270; Fig 4D, E; Fig 5.

Other changes are marked in red.

Thank you for your comments that help us a lot.

---

## [Decision Letter · Decision Letter 1]

16 May 2023

Negative regulation of angiogenesis and the MAPK pathway may be a shared biological pathway between IS and epilepsy.

PONE-D-23-03945R1

Dear Dr. Gong,

We’re pleased to inform you that your manuscript has been judged scientifically suitable for publication and will be formally accepted for publication once it meets all outstanding technical requirements.

Kind regards,

Yuzhen Xu

Academic Editor

PLOS ONE

Additional Editor Comments (optional):

Reviewers' comments:

Reviewer's Responses to Questions

**Comments to the Author**

1. If the authors have adequately addressed your comments raised in a previous round of review and you feel that this manuscript is now acceptable for publication, you may indicate that here to bypass the “Comments to the Author” section, enter your conflict of interest statement in the “Confidential to Editor” section, and submit your "Accept" recommendation.

Reviewer #2: All comments have been addressed

2. Is the manuscript technically sound, and do the data support the conclusions?

Reviewer #2: Yes

3. Has the statistical analysis been performed appropriately and rigorously? 

Reviewer #2: Yes

4. Have the authors made all data underlying the findings in their manuscript fully available?

Reviewer #2: Yes

5. Is the manuscript presented in an intelligible fashion and written in standard English?

Reviewer #2: Yes

6. Review Comments to the Author

Reviewer #2: Authors have addressed all my concerns. I have no further comments and suggest the next step towards publication.

7. PLOS authors have the option to publish the peer review history of their article (what does this mean?). If published, this will include your full peer review and any attached files.

Reviewer #2: No

---

## [Editor Report · Acceptance letter]

18 May 2023

PONE-D-23-03945R1 

Negative regulation of angiogenesis and the MAPK pathway may be a shared biological pathway between IS and epilepsy 

Dear Dr. Gong:

I'm pleased to inform you that your manuscript has been deemed suitable for publication in PLOS ONE. Congratulations! Your manuscript is now with our production department. 

Kind regards, 

on behalf of

Professor Yuzhen Xu 

Academic Editor

PLOS ONE